# Biodegradable Biocomposite of Starch Films Cross-Linked with Polyethylene Glycol Diglycidyl Ether and Reinforced by Microfibrillated Cellulose

**DOI:** 10.3390/polym16091290

**Published:** 2024-05-04

**Authors:** María M. González-Pérez, María G. Lomelí-Ramírez, Jorge R. Robledo-Ortiz, José A. Silva-Guzmán, Ricardo Manríquez-González

**Affiliations:** Department of Wood, Cellulose and Paper, University Center for Exact Sciences and Engineering, University of Guadalajara, km 15.5 at the Guadalajara-Nogales Highway, Zapopan 45220, Mexico; maria.gonzalez3738@alumnos.udg.mx (M.M.G.-P.); jorge.robledo@academicos.udg.mx (J.R.R.-O.); jantonio.silva@academicos.udg.mx (J.A.S.-G.)

**Keywords:** films, biocomposite, microfibrillated cellulose, OCC cardboard, PEGDE cross-linking agent

## Abstract

Biopolymers are biodegradable and renewable and can significantly reduce environmental impacts. For this reason, biocomposites based on a plasticized starch and cross-linker matrix and with a microfibrillated OCC cardboard cellulose reinforcement were developed. Biocomposites were prepared by suspension casting with varied amounts of microfibrillated cellulose: 0, 4, 8, and 12 wt%. Polyethylene glycol diglycidyl ether (PEGDE) was used as a cross-linking, water-soluble, and non-toxic agent. Microfibrillated cellulose (MFC) from OCC cardboard showed appropriate properties and potential for good performance as a reinforcement. In general, microfiber incorporation and matrix cross-linking increased crystallization, reduced water adsorption, and improved the physical and tensile properties of the plasticized starch. Biocomposites cross-linked with PEGDE and reinforced with 12 wt% MFC showed the best properties. The chemical and structural changes induced by the cross-linking of starch chains and MFC reinforcement were confirmed by FTIR, NMR, and XRD. Biodegradation higher than 80% was achieved for most biocomposites in 15 days of laboratory compost.

## 1. Introduction

Plastic waste management is a global environmental concern that requires innovative and sustainable solutions. Applying practices and regulations to reduce the ecological footprint of plastic waste requires close collaboration between the public and private sectors, as well as greater education and environmental awareness.

Bioplastics and biopolymers are promising alternatives to conventional plastics, but their environmental impact depends on their production, use, and disposal. Biodegradability and compostability are important characteristics of materials used in environmental applications, and the degradation mechanism of materials depends on their chemical composition and molecular structure. Biodegradable materials based on biopolymers have been developed as an alternative to traditional plastics in response to the growing environmental awareness of reducing non-biodegradable plastic waste [1]. Starch is one of the most promising natural polymers for the development of these materials due to its renewability, total degradability, high availability, and the diversity of botanical sources from which it can be obtained [2,3,4].

However, these materials have some disadvantages, such as their highly hydrophilic character, reduced mechanical properties, and low thermal stability compared to conventional polymers [5,6]. In this regard, several methods have been proposed to improve the mechanical properties of starch-based biodegradable materials, such as combination with other biodegradable polymers, cross-linking, and reinforcement with mineral particles or cellulosic fibers [7,8,9,10,11]. Incorporating natural fibers in composite materials can improve their mechanical properties and contribute to their biodegradability, which is important to reduce the environmental impact of plastics [12,13].

On the other hand, microfibrillated cellulose (MFC) is attractive as a reinforcing material due to its high strength, low density, and large surface area [14,15]. Using MFC as a reinforcing material for starch films has significantly enhanced various physical, mechanical, and surface properties. These include improved resistance to water absorption, greater thermal stability, and enhanced barrier properties and permeability, among others [16,17,18]. Furthermore, starch cross-linking involves the interaction between the amylose and amylopectin chains, forming a three-dimensional network structure that can be carried out through covalent bonds [19]. However, it has been reported that most compounds commonly used for this process are toxic, expensive, and inefficient [5,20]. In this sense, the present work proposes using PEGDE as a non-toxic and cost-effective cross-linking agent to improve the mechanical properties of the biocomposite and its resistance to water absorption [21,22,23]. Biodegradability is another important aspect of plastics developed from starch, so it is of interest to evaluate their environmental impact and how their decomposition can be influenced by factors such as temperature, humidity, and the presence of microorganisms [4].

Thus, the present research aimed to develop biodegradable biocomposite films with attractive physical and mechanical properties, wet adsorption resistance, and low-cost production. All these characteristics could be achieved with a sustainable vision of the biocomposite including (1) a renewable biopolymer available as starch, (2) a chemically safe glycerine plasticizer and PEGDE cross-linker, and (3) a reinforcement material composed of cellulose microfibers of a greater surface area from recycled OCC cardboard sheets. The biocomposite films were characterized spectroscopically by FTIR, solid-state NMR, and XRD. Their mechanical and morphological properties were evaluated by tensile tests and SEM, respectively. The water permeability of the biocomposite films was determined by a moisture adsorption test. Finally, a preliminary biodegradability test was carried out following a procedure adapted to the ISO 20200 [24] standard for disintegrating plastic materials in laboratory composting.

## 2. Materials and Methods

### 2.1. Materials

Corn starch was supplied by IMSA (Industrializadora de Almidón de Maíz S.A. de C.V.), Mexico. Then, 99% glycerol was purchased from Golden Bell, Mexico, and poly (ethylene glycol) diglycidyl ether (PEGDE) M_n_ 500 was purchased from Sigma-Aldrich, Toluca, Mexico. Microfibrillated cellulose was obtained from old container corrugated (OCC) cardboard sheets supplied by International Papers, USA.

### 2.2. Methods

#### 2.2.1. Microfibrillated Cellulose (MFC)

Recycled OCC cardboard sheets were used as a source of lignocellulosic fiber to obtain microfibrillated cellulose (MFC), which was used as the reinforcing material for the starch films. The microfibrillated cellulose (unbleached) was only treated by an alkaline pulping process under mild conditions to remove some impurities and reduce the lignin content. The alkaline pulping process was carried out using a Jayme digester, where native OCC fibers were combined with a pulping liquor consisting of 10% NaOH and 0.1% anthraquinone. The production of the MFCs was carried out through a mechanical fibrillation process using a Super Mascolloider Microprocessor colloidal mill (Masuko Sangyo MKCA6-2). Further information on this methodology can be found in the work of González et al. [25].

#### 2.2.2. Biocomposites Films of Starch–PEGDE–MFC

The ratio of glycerol, cross-linker agent, and reinforcing fibers in the starch was preliminary tested to determine the appropriate concentration of each component to obtain biocomposite films with optimal physical–mechanical properties. Thus, biocomposites were developed in film form from starch plasticized with 30% glycerol (PS), cross-linked using 3% PEGDE (CS), and reinforced with microfibrillated cellulose (MFC) in compositions ranging from 4 to 8 and 12% *w*/*w* (starch-based), see Table 1. Figure 1 presents the methodology and conditions used to produce the films using the casting technique. Figure 2 shows some images of the obtained starch films, which were dimensionally stable, free of bubbles, and translucent. Table 1 shows the codification and composition of the produced biocomposite starch films and the control samples.

#### 2.2.3. Physical–Mechanical Performance and Water Adsorption Evaluation

*Physical–Mechanical Performance.* The tensile properties were determined following the ASTM D882-02 [26] standard using the universal testing machine Instron 3345. The specimens were cut into dimensions of 10 mm width × 170 mm length × 0.217 ± 0.027 mm thickness and conditioned at 50% relative humidity for 48 h. The initial grip separation was 100 mm, using a cell of 1 kN and a speed of 25 mm/min. A minimum of five samples were tested to obtain average values of tensile strength, Young’s modulus, and elongation at break. An analysis of variance (ANOVA) and Tukey test (*p* < 0.05) were carried out to determine significant differences.

*Moisture absorption.* A moisture absorption study was carried out according to ASTM E104-02 [27], which establishes the standard method to maintain constant relative humidity using aqueous solutions. A hermetic glass container of 38 × 30 × 24 cm^3^ was used and equipped with a fan to ensure humidity homogeneity. The test was carried out at a relative humidity of 75% using a saturated NaCl solution. Samples of 25 × 25 mm^2^ × 0.217 ± 0.027 mm were dried to a constant weight in a conventional laboratory oven (Luzeren DHG-9070A) at 30 °C for 72 h and then at 60 °C for 1 h.

#### 2.2.4. Morphological Characterization

*Scanning Electron Microscopy (SEM).* The fracture zone of the specimens from the tensile test was observed by scanning electron microscopy using a Tescan MIRA 3 LMU microscope. The samples were placed on a sample holder using carbon tape and coated with gold for 20 s. All samples were measured with a voltage of 15 kV.

#### 2.2.5. Chemical and Structural Characterization

*Fourier Transform Infrared Spectroscopy (FTIR).* Structural and chemical changes of the biocomposite films due to the cross-linking of starch and the incorporation of microfibers were measured by FTIR analyses using an ATR device. All FTIR spectra were obtained with 16 scans in a spectral window of 4000 to 700 cm^−1^ (Perkin Elmer Spectrum GX) and a resolution gap of 4 cm^−1^. 

*Solid-State Nuclear Magnetic Resonance (ssNMR).* The chemical changes and cross-linking of biocomposites were evaluated by a solid-state NMR technique using a ^13^C CPMAS experiment. All NMR measurements were performed with a Jeol ECA 600 spectrometer of 14 Teslas. In total, 50 mg of each sample (prior powdering) was introduced into a 4 mm Si_3_N_4_ rotor and analyzed with a two-channel (H, X) DOTY solid-state probe at room temperature at a spinning speed of 10 KHz. A ^13^C CPMAS experiment was employed to measure the carbon nuclei of the samples using 150.9 MHz of operating frequency with a 90° pulse width of about 3.08 μs, and the ^13^C CP was acquired with 2 ms of contact time and 5 s of acquisition delay.

*Crystallinity by X-ray Diffraction (XRD).* The crystallinity degree of the films was calculated following a method previously described [28,29,30], based on a comparison between the area of the crystalline zone ($A_{c}$) with the total area under the diffraction pattern, which included the peaks of the crystalline zone and the halo of the amorphous zone ($A_{a}$). Origin Pro 8.5 software was used to plot a curve based on a Gaussian model to calculate the areas and determine the crystallinity degree ($X_{c})$ according to Equation (1).
(1)$$X_{c}\left(\%\right)=\frac{A_{c}}{A_{c}+A_{a}}*100$$

#### 2.2.6. Preliminary Study of Biodegradability

A preliminary study of the biodegradability of biofilms was carried out following an adaptation of the ISO 20200 standard [24], which establishes the method to evaluate the disintegration of plastic materials in laboratory composting. The synthetic compost was composed of 40% sawdust, 30% rabbit feed, 10% corn starch, 7% organic compost, 3% chicken manure, 5% sucrose, 4% corn seed oil, and 1% urea. Samples of 25 × 25 mm^2^ × 0.217 ± 0.027 mm were dried at 30 °C for 72 h and then at 60 °C for 60 min. The initial weight was recorded, and the samples were conditioned at 50% relative humidity for 5 days in polyethylene mesh bags. A triplicate experiment was carried out in containers with 34 g of compost each. The weight of the containers was recorded, and they were incubated at 58 °C for 15 days in a laboratory oven (LUZEREN DHG-9070A). Finally, the compost was dried at 105 °C, and the percentage of dried solids was determined.

## 3. Results and Discussion

### 3.1. Physical and Mechanical Tests

#### 3.1.1. Physical–Mechanical Performance

Table 2 shows the results of the tensile tests of the films, taking PS as a reference, whose tensile strength was 2 MPa. The effect of starch cross-linking and the incorporation of OCC cardboard MFC increased the resistance of the materials.

In the case of non-cross-linked composite films, a tendency toward increased tensile strength was observed as the reinforcement content increased. The highest resistance value was obtained with 12% MFC (13.0 MPa). The Young’s modulus also increased with the reinforcement content, presenting the highest values with 12% MFC (212.7 MPa). Similar results have been reported in the literature [6,31,32,33].

Incorporating PEGDE as a cross-linking agent increased the tensile strength and Young’s modulus as the reinforcement content increased, with the highest values obtained at 12% MFC (14.3 MPa and 189.3 MPa, respectively). PEGDE has a chemical structure that provides more degrees of freedom for interaction. It also has two epoxy groups at its ends, allowing it to react with the OH groups in the starch and cellulose, producing an ether bond (detected by NMR). Kiuchi et al. [21] and Xiao et al. [34] reported that the chain length of PEGDE was a relevant factor when cross-linking polysaccharides, having an important effect on the mechanical performance of the materials studied.

In general, the composite and cross-linked films presented less deformation. From the values obtained, it is possible to point out that the incorporation of microfibrillated cellulose as a reinforcing material and the chemical cross-linking of the amylose and amylopectin or even cellulose chains gave a material with greater rigidity. This can be attributed to the contraction of the chains comprising the matrix, along with a good reinforcement distribution that contributed to the formation of a more compact network with stronger interactions, which reduced the material deformation when subjected to tensile stresses. Similar results using other compounds as cross-linkers have been reported by Ghosh and Netravali [35], Guleria et al. [36], and Patil and Netravali [6].

#### 3.1.2. Moisture Absorption

All the materials evaluated presented equilibrium after 60 h of conditioning in an environment at 75% controlled relative humidity (RH) (Figure 3a). With the incorporation of MFC and cross-linking, the resistance to moisture absorption increased compared to PS. For the biocomposite films with 12% reinforcement (BM12), a reduction in moisture absorption greater than 15% was obtained, while the films with 12% MFC and cross-links (CBM12) showed reductions of 19%. This effect was because the functional groups on starch and cellulose surfaces resulted in good interfacial adhesion between both molecules, improving the water resistance and mechanical properties. Additionally, microfibers are less hygroscopic than starch because they have a higher degree of molecular order [37] and, since they are not bleached, they have a presence of lignin and hemicelluloses, components with greater hydrophobicity than cellulose [38,39].

The diffusion coefficients (D) were determined from moisture uptake data using a hindered diffusion model proposed by Carter and Kibler [40], which resembles a Lang-muir-type model for anomalous moisture diffusion. This model, successfully applied to various composite materials [40,41], links absorption to free-volume availability and polymer–water affinity. Besides the diffusion coefficient, the model has two additional parameters, as observed in Equation (2), that describe the probability that free moisture molecules will bind (β) or release (α) bound molecules.
(2)$$\frac{M_{t}}{M_{\infty }}=1-\frac{\beta }{\beta +\alpha }e^{-\alpha t}-\frac{\alpha }{\left(\beta +\alpha \right)}\frac{8}{\pi^{2}}e^{\frac{-{D\pi }^{2}t}{l^{2}}}$$
where $M_{t}\;$ is the moisture absorbed at any time, $M_{\infty }\;$ is the moisture absorbed at equilibrium, D  is the diffusivity coefficient, and l is the film thickness. Table 3 displays the parameters calculated via non-linear regression using Matlab^®^. It was noted that while the diffusivity coefficient was higher for PS, the maximum water uptake decreased due to the presence of MFC. The curves constructed based on the data obtained with the model for PS and CBM12 are shown in Figure 3b.

### 3.2. Morphological Characterization

#### Scanning Electron Microscopy (SEM)

Using the scanning electron microscopy technique, a morphological characterization of the films was applied to observe the interactions and distributions of the materials. The fracture zones of the tensile test specimens of PS, CS, and biocomposite films with the highest reinforcement content (12%), as well as starch granules and MFC, were examined. In Figure 4a, the morphology of the native corn starch granules can be observed, exhibiting a polyhedral and irregular shape. These micrographs coincide with what has been reported in the literature for this type of starch [42,43]. Figure 4b shows the micrograph of the plasticized starch, where it can be observed the morphology of the matrix, as well as the absence of starch granules (4a) after the destructuring and plasticization process, showed a smooth surface that coincides with those reported by Cuevas-Carballo [30]. In Figure 4c, which corresponds to the cross-linked starch film CS, a homogeneous surface is observed that corresponds to a continuous phase, and, when compared against PS, differences can be noted in the morphology of the fracture zone. Furthermore, CS presented smoother surfaces free of pores and cracks, characteristic of materials that present a type of brittle failure, while, in the fracture zone of PS, a deformation with greater reliefs was observed. These images agreed with the tension test results, where PS presented a type of ductile failure [31].

On the other hand, Figure 5 shows micrographs with a change in the morphology of the films when incorporating 12% reinforcement to the plasticized starch matrix BM12 (a) and the cross-linked matrix CBM12 (b). Here, both show a rougher surface with some cavities, in which a good distribution of the reinforcement could be distinguished without agglomerates.

By zooming in on the fracture zone of the CBM12 film (Figure 6a,b), some microfibers embedded in the cross-linked starch matrix and the good matrix–reinforcement interaction could be seen. Figure 6c shows only the MFC, confirming that the dimensions of the microfibers scale ranged.

### 3.3. Chemical and Structural Characterization

#### 3.3.1. FTIR Analysis

FTIR-ATR infrared spectroscopy was performed to evaluate the chemical and structural changes of the films due to the chemical cross-linking of starch and the incorporation of microfibers (Figure 7). The spectra of films with plasticized starch (PS), cross-linked starch (CS), and biocomposite with 12% MFC reinforcement were analyzed as a comparison. The spectrum of PS showed a typical signal pattern of starch, with a broad band at 3300 cm^−1^ attributed to OH stretching vibrations from hydroxy groups. C-H stretching vibration of the aliphatic part of the anhydroglucose repeat units (AGUs) in the polysaccharide was observed between 2880 and 2900 cm^−1^ and was corroborated by the C-H bending vibration in the region from 1500 to 1300 cm^−1^. The signal at 1635 cm^−1^ was due to adsorbed water in the sample, which is typical in polyhydroxylated polymers. Finally, the intense signal at 1050 cm^−1^ was attributed to the C-O stretching vibration of the alcohol (C-OH) and ether (C-O-C) functional groups [4,35,44,45,46,47].

Regarding the cross-linking of the matrix, some bands were slightly more intense in the OH signal of the PS compared with the CS. This difference can be attributed to the new ether formation in the PEG cross-linking that took the available OH groups from the starch. However, samples with MFC (BM12 and CBM12) depicted intense OH signals due to the OH groups from the cellulose. In contrast, the C-O signal intensity in the CS sample was higher than in PS due to the clear contribution of the new ether groups from the PEG cross-linker. This was also observed in BM12 and CBM12 biocomposites by the presence of the C-O groups from cellulose (MFC). On the other hand, a new small signal at 1723 cm^−1^ in CS and BM12 samples was attributed to the C=O stretching vibration of carbonyl groups, which almost disappeared in the CBM12 biocomposite. This group could be formed by a slight decomposition of the starch after the thermal treatment in the film formation, as already reported [4,47]. In the case of CBM12, the C=O signal intensity was reduced due to the incorporation of microfibrillated cellulose, which diluted the presence of these carbonyl groups.

#### 3.3.2. ^13^C CPMAS NMR Analysis

As the FTIR spectra were only able to observe small differences in the films of CS and PS, solid-state NMR ^13^C was employed to obtain more information about the presence of glycerol (PS) and the cross-linking PEGDE (CS) in the starch films. Figure 8 shows the ^13^C NMR spectra of plasticized starch (a) and cross-linked starch (b) films. 

In both spectra of Figure 8, the six carbon signals from the AGU repeat units (α-D-glucopyranose) of the starch were clearly observed between 104 and 60 ppm. The signal of C1 was located from 104 to 100 ppm, that of C4 at 82 ppm, the most intense signal between 80 and 70 ppm was attributed to C2,3 and 5, and the C6 signal was observed at 62 ppm [48]. Furthermore, the spectrum of Figure 8 shows a new signal at 64 ppm that corresponded to the tertiary carbon from the glycerol used as a plasticizer. 

The ^13^C NMR spectrum of starch with cross-linking PEGDE (CS) is shown in Figure 8. The PEGDE presence in the sample was confirmed by the carbon signals at 72 and 61 ppm. The first was attributed to the methine connector (Cb) from the PEG cross-linking agent after the ether bond formation with one hydroxyl group (C6, C2, or C3) from the starch. The second was associated with methylene carbons in ether moieties (CH2-O-), possibly from the PEG repeat unit chain in the cross-linking agent (Ca, Cc, and Cd). Similar results were obtained by Kono [49] cross-linking carboxymethylcellulose (CMC) with PEGDE. Furthermore, the signal intensity of C1 at 104 ppm represented an increase in the amorphous structure of the starch due to the presence of glycerol (PS) and cross-linking by PEG (CS) [50]. However, in this CS film, the intensity of this signal was even higher than in the PS film, which may have been because cross-linking had a greater impact on the loss of crystallinity of the starch or because of the lower amount of water in the CS sample [51]. This was also in agreement with the moisture adsorption test for these films.

#### 3.3.3. X-ray Diffraction Analysis (XRD)

The crystallinity of native fibers and MFC was documented in a previous publication [25]. The calculated relative crystallinity index (Cr.I.) showed values ranging from 80 to 85, with the highest values observed in the microfibrillated cellulose.

Changes in the crystallinity of the films were evaluated as an effect of destructuring, cross-linking, and the incorporation of the MFC reinforcement through X-ray diffraction analysis. Figure 9 shows the crystallographic patterns obtained for the films and their references. Firstly, it is possible to observe that the native starch showed a type A pattern typical of cereals, which has been reported to be characteristic of corn starch, presenting the four distinctive peaks, two individual peaks at 15° and 23° and a doublet between approximately 17° and 18° [42,52,53].

The crystallographic pattern of the plasticized starch (PS) showed differences with respect to the pattern of native starch; this is because, during the gelatinization stage, destructuring of the granules was achieved, which allowed the amylose and amylopectin chains to take positions with a greater number of degrees of freedom. This was due to the glycerin, which reduced compaction and maintained this new structural conformation in the more amorphous material [54]. New crystallographic patterns E_h_, V_h_, and V_a_ appeared at 16.5°, 19.5°, and 21.5°, respectively, which are clearly defined and have been previously reported in the literature by Van Soest et al. [53] and Montero et al. [55], who mentioned in their work that the appearance of these new crystallographic patterns is due to two effects: (1) the residual crystallization of amylose, which can occur due to the incomplete destructuring of the starch granules, maintaining some original crystalline areas, and (2) the recrystallization of amylose, which may originate during processing and plasticization. Here, the linear amylose chains could form complexes with the plasticizer through interactions with their OH groups, presenting new single-helix crystal structures. According to Van Soest et al. and Montero et al., the V_h_-type crystallographic pattern, which corresponds to a hydrated structure formed by six amylose helices, can be transformed into the V_a_-type crystallographic pattern under dehydration conditions, and becomes an E_h_ conformation when the retrogradation or aging process occurs in the plasticized starch. This new structure (E_h_) comprises seven amylose helices and is only stable without humidity.

Among the works that stand out in the literature referring to this topic are those carried out by Lomelí-Ramírez et al., Zobel, Van Soest et al., and Montero et al. [4,52,53,55], where starches that have been subjected to gelatinization and plasticization processes were evaluated. They reported that the appearance of these structures, as well as the intensity they display, depend on the nature of the starch, the plasticizer used and the amount of it, the processing method and parameters, and the humidity conditions. Similar effects were obtained for the cross-linked films, where the appearances of the three crystallographic patterns in the same regions with slight differences in definition were shown. This seems to be opposite to the NMR results of the CS sample with a higher degree of amorphous component than the PS film. However, it has been reported that NMR analysis can evaluate helix content by detecting short-rage structure order, whereas XRD detects long-rage crystallographic order by measuring the crystalline domain [51,56]. 

For the materials that had MFC in their composition, the crystallographic patterns E_h_ and V_h_ appeared as well as a new peak at approximately 22.5°, which corresponded to the presence of cellulose in the starch films because this peak appeared in an area very close to that of V_a_ (21.5°). This V_a_ peak could have overlapped with the cellulose peak, thus reducing its visibility [55,57]. Some reported studies in which starch biocomposites with the reinforcement of lignocellulosic materials were evaluated obtained similar results [4,47,55,58]. The same trend was observed for the material with reinforcement and cross-linking (CBM12), with some differences in the width and definition of the peaks.

Table 4 shows the values of the crystallinity degree ($X_{c})$ of the films with 12% reinforcement as well as the reference materials, which were determined according to a method reported in the literature by Nara et al., Frost et al., and Cuevas-Carballo [28,29,30], which is based on the relationship that exists between the area that represents the crystalline zone (peaks) and the total area under the diffraction pattern (peaks of the crystalline zone plus the halo of the amorphous zone). From this point, it was possible to establish a trend in the crystallographic behavior of the films associated with cross-linking and reinforcement and its direct relationship with the physical and mechanical properties of the materials. It was found that through the gelatinization and plasticization of native starch, a reduction in the crystallinity degree of 68% was obtained, resulting from the destructuring of the granular conformation of the amylose chains and amylopectin. Some authors have reported values of the degree of crystallinity of plasticized starch between 10% and 15%, the variation of which depended on the amylose content, the order of the internal structure, and the processing method [59].

Through chemical cross-linking with PEGDE, an increase in the crystallinity degree was obtained. This effect may have been due to the chemical nature of PEGDE, which has a longer chain and provides more degrees of freedom that allow for greater mobility of the chains when forming the network, thus achieving a more ordered material with fewer defects [21]. The incorporation of MFC also contributed to increasing the crystallinity degree of the films. Ilyas et al. [60] reported values of the crystallinity degree of biocomposites with nanofibrillated cellulose reinforcement, finding a tendency to increase crystallinity. This is because introducing a material with greater crystallinity gives more crystalline areas to the biocomposite.

### 3.4. Biodegradability of the Biocomposites

#### Preliminary Biodegradability Test

The preliminary biodegradability test was based on the determination of the disintegration degree of the materials in a laboratory test, with simulated conditions of the aerobic composting process, with a gravimetric recording of the materials at the beginning and end of the test. Figure 10 shows images from the test, where the mycelium was present in the compost from the first week of incubation and, as the days progressed, the mycelium spread until it covered a large part of the area of the container, and the test was carried out. The incubation period was stopped on day 15 because the PS and CS films had already disintegrated almost completely and were difficult to handle. The dry solids (Ds) corresponding to the compost used obtained at the end of the test were 45.5%.

Once the incubation period ended, the rest of the material was weighed, and the results of the disintegration degree were obtained, which were associated with the weight loss and are reflected in Table 5.

Most of the materials presented disintegration percentages above 80%. It was observed that the incorporation of reinforcement helped to reduce the disintegration of PS in the period evaluated. Films with the highest reinforcement content presented the lowest degradation. Regarding cross-linking with PEGDE, it was observed that the values were also reduced compared to PS. The results found in this study suggest that the cross-linking of starch and the incorporation of reinforcement did not interfere with the biodegradation of the materials.

According to Azwa et al. [61] the degradation of biocomposites is influenced by factors such as humidity, temperature, and the enzymatic activities of microorganisms. Similar results have reported the biodegradability of starch films by composting. Torres et al. [62] evaluated the biodegradability of starch films from different botanical sources for an incubation period of 30 days, finding percentages of loss greater than 90% in all cases, with the highest found for cassava starch (99.35%) and the lowest found for potato starch (90.03%). González-Seligra et al. [63] evaluated the biodegradation of starch-based films cross-linked with citric acid, finding that the cross-linked films biodegraded in a longer time (18 days) than the non-cross-linked ones (12 days). This was attributed to the fact that the incorporation of citric acid decreased moisture absorption, reducing the microbial attack on the samples. In this work, PS and CS showed the highest degradation values during the evaluated period, whereas the biocomposite with 12% reinforcement (BM12) showed the highest resistance to degradation.

## 4. Conclusions

Corn starch biocomposite films were prepared by adding glycerol (PS), PEGDE (as a cross-linking agent) (CS), and MCF of OCC reinforcement in different contents (4–12%) with and without cross-linking. The mechanical, water adsorption, and biodegradation properties were tested and associated with the preparation and composition of the resulting biocomposites. 

All the developed materials presented better mechanical resistance properties and greater resistance to moisture absorption when compared to the reference material (PS). PEGDE cross-linked plasticized starch films with a 12% reinforcement of MFC of OCC cardboard showed the best results for tensile strength and tensile modulus as well as increased water adsorption resistance by up to 19% in comparison with the PS film. As could be expected, the hydrophilic character of the plasticized starch was decreased with cross-linking and reinforcement. The moisture absorption of the studied materials followed the Langmuir diffusion model.

Spectroscopic characterization by FTIR and ^13^C solid-state NMR demonstrated structural and chemical changes in the obtained biocomposite films. Changes in signal intensity or signal patterns in the samples determined the presence or absence of PEGDE and MFC in the biocomposite films. These changes due to the chemical and structural contribution of the cross-linking agent and the reinforcing fibers were clearly associated with the improvement of the properties of the biocomposite films. Furthermore, ^13^C NMR analysis showed an amorphous increase of the starch structure after cross-linking, associated with the higher C1 signal intensity of the AGU repeat unit. In contrast, the measurement by XRD showed the destructuring and plasticization of the native starch (type A) giving rise to new crystallographic patterns (E_h_, V_h_, and V_a_), attributed to the recrystallization of amylose. The highest crystallinity percentages were obtained for the biocomposites with the highest reinforcement content, associated with the cellulose crystalline domain. Furthermore, in the preliminary biodegradation test of the materials studied in compost, it was found that the incorporation of MFC had a more significant effect in reducing weight loss at the end of the incubation period. After 15 days, the PS and CS films were almost disintegrated and the BM and CBM samples achieved disintegration percentages higher than 80%.

Finally, important outcomes can be highlighted regarding the importance of the cross-linker agent’s contribution to the starch film’s stability as well as the reinforcing network of microfibrillated cellulose that offered more available interaction contact area for the starch. In addition, the equilibrated combination of starch, PE, GDE, and MFC contributed to the improvement of the mechanical and water-resistant properties of the materials, including adequate biodegradation time, as was demonstrated in the CBM12 biocomposite. Thus, the present biocomposite proposal shows an interesting alternative to replace conventional materials or composites formed by non-biodegradable plastic polymers but with competitive mechanical and water-resistant properties.

## Figures and Tables

**Figure 1 polymers-16-01290-f001:**
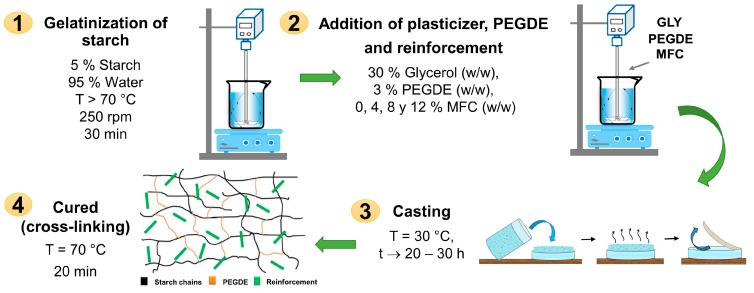
Biocomposite formation via casting technique.

**Figure 2 polymers-16-01290-f002:**
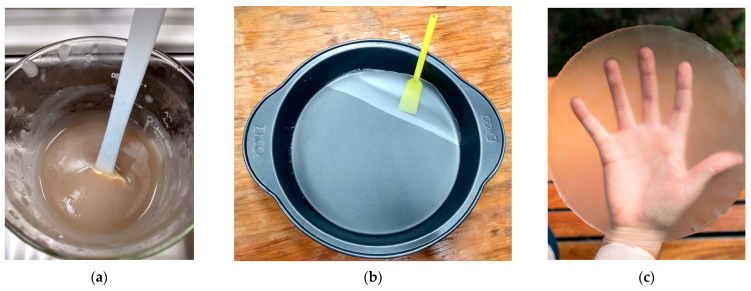
Biocomposite of starch–PEGDE–MFC. (**a**) Mixture of starch–PEGDE–MFC. (**b**) Film release formed by casting technique. (**c**) Cross-linked starch film with 12% MFC.

**Figure 3 polymers-16-01290-f003:**
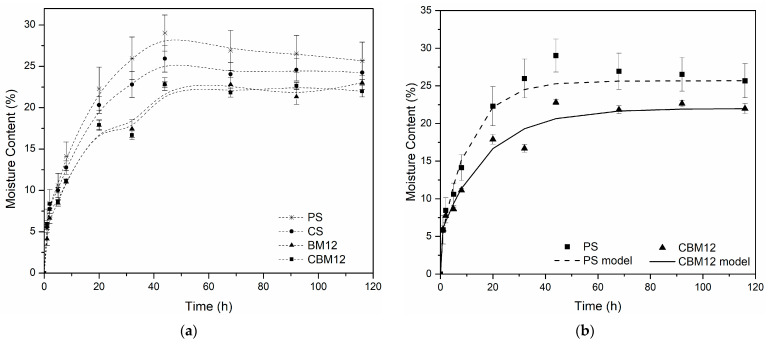
Moisture absorption curves. (**a**) Experimental moisture absorption data of the films with the highest MFC content and their references. (**b**) Comparison of experimental data (points) against the model developed from the Langmuir diffusion equation (lines).

**Figure 4 polymers-16-01290-f004:**
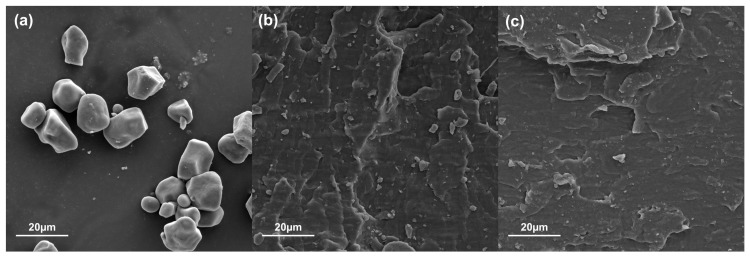
Starch granules (**a**), PS (**b**), CS (**c**).

**Figure 5 polymers-16-01290-f005:**
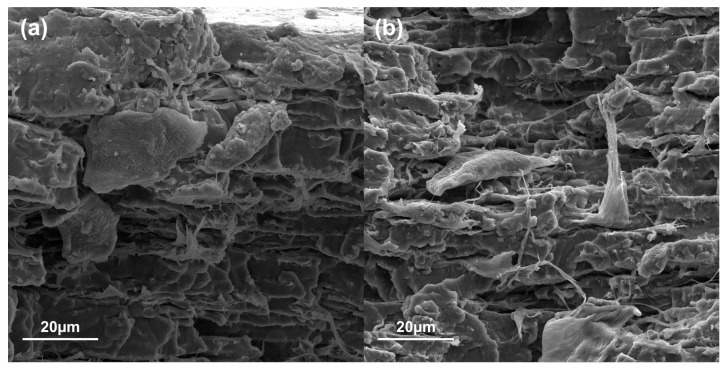
Biocomposite films BM12 (**a**) and CBM12 (**b**).

**Figure 6 polymers-16-01290-f006:**
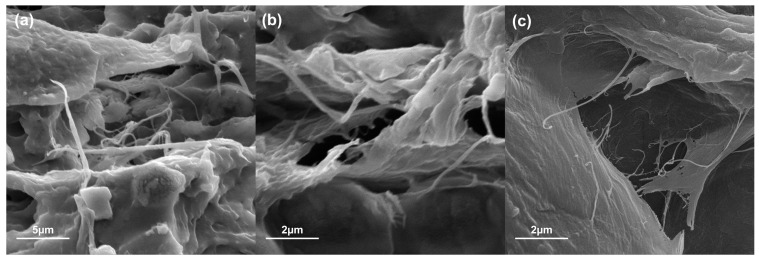
Interaction of the matrix and the reinforcing material of CBM12 magnified images (**a**,**b**) and MFC (**c**).

**Figure 7 polymers-16-01290-f007:**
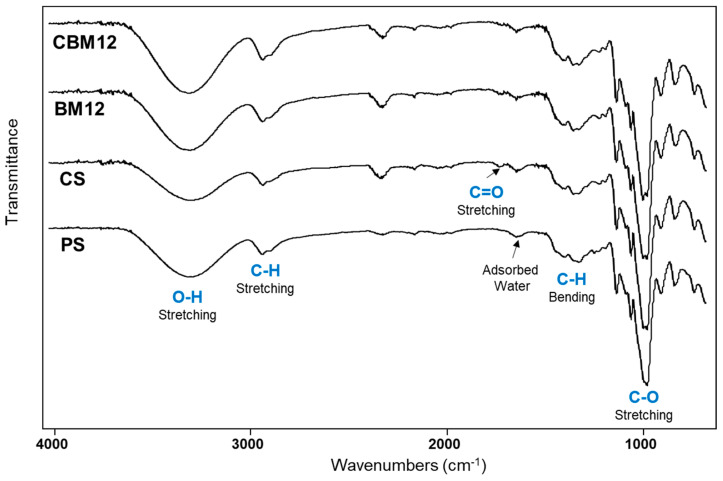
FTIR spectra of PS, CS, and plasticized and cross-linked biocomposite films with 12% MFC (BM12 and CBM12).

**Figure 8 polymers-16-01290-f008:**
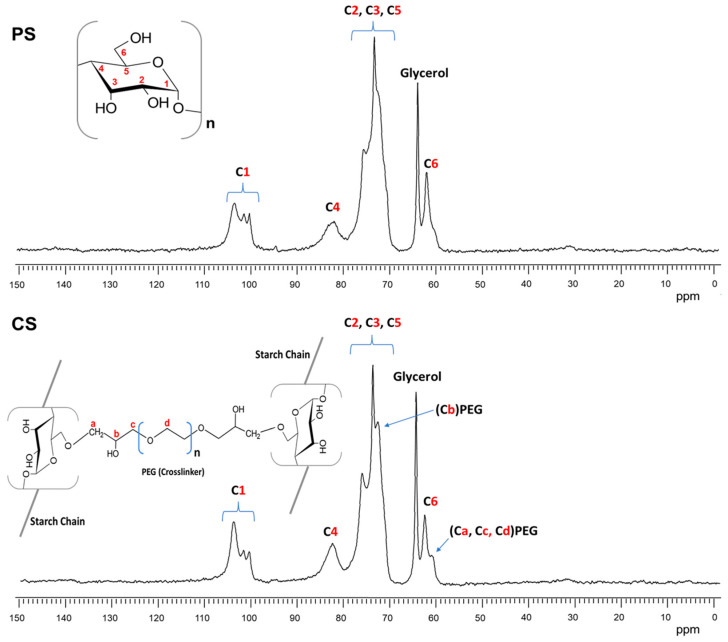
^13^C NMR CPMAS spectra of plasticized starch (PS) and cross-linked starch (CS).

**Figure 9 polymers-16-01290-f009:**
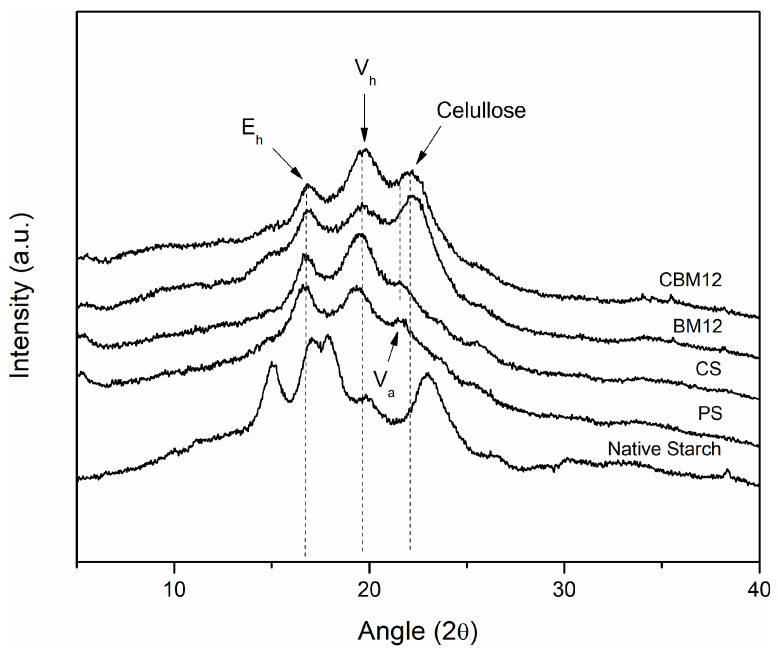
Crystallographic patterns of cross-linked biocomposites and their reference materials.

**Figure 10 polymers-16-01290-f010:**
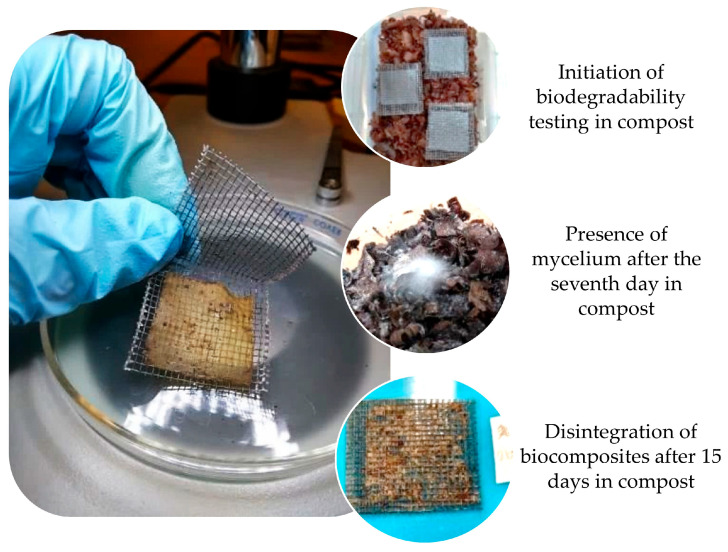
Preliminary test of biodegradability.

**Table 1 polymers-16-01290-t001:** Film compositions and code names.

Code Name	Sample
PS	Plasticized starch film ^1^
CS	Cross-linked starch film ^1^
BM4	Biocomposite film with 4% microfibrillated cellulose
BM8	Biocomposite film with 8% microfibrillated cellulose
BM12	Biocomposite film with 12% microfibrillated cellulose
CBM4	Cross-linked biocomposite film with 4% microfibrillated cellulose
CBM8	Cross-linked biocomposite film with 8% microfibrillated cellulose
CBM12	Cross-linked biocomposite film with 12% microfibrillated cellulose

^1^ PS and CS had 0% reinforcement content.

**Table 2 polymers-16-01290-t002:** Tensile properties of starch films.

Sample	Tensile Strength (MPa)	Young’s Modulus (MPa)	Elongation at Break (%)
PS	2.0 ± 0.1 ^a^	5.8 ± 0.8 ^a^	76.8 ± 9.6
CS	4.2 ± 0.8 ^b^	26.4 ± 4.3 ^b^	45.9 ± 9.7
BM4	7.7 ± 0.8 ^c^	60.4 ± 11.7 ^c^	32.7 ± 3.4
BM8	11.2 ± 1.4 ^d^	130.5 ± 21.5 ^d^	21.0 ± 3.9
BM12	13.0 ± 1.7 ^d^	212.7 ± 52.3 ^e^	17.9 ± 3.7
CBM4	8.4 ± 1.2 ^c,e^	57.1 ± 9.9 ^c^	34.8 ± 6.9
CBM8	9.2 ± 0.9 ^e^	83.4 ± 7.4 ^f^	22.5 ± 5.8
CBM12	14.3 ± 3.0 ^d^	189.3 ± 32.0 ^e^	21.1 ± 3.4

The letters a–f indicate significant differences (*p* < 0.05).

**Table 3 polymers-16-01290-t003:** Calculated parameters from the Langmuir diffusion model.

Sample	$M_{\infty }\;$ (%)	D (10^10^ m^2^/s)	α (1/s)	β (1/s)
PS	26.9	0.5499	0.09223	0.5502
BM12	22.8	0.7317	0.0566	0.2801
CBM12	21.8	0.7513	0.06513	1.6890

**Table 4 polymers-16-01290-t004:** Crystallinity degree of native starch, films, and biocomposites.

Sample	$X_{c}\;(\%)$
Native Starch	39.0
PS	12.5
CS	14.6
BM12	12.7
CBM12	14.8

**Table 5 polymers-16-01290-t005:** Disintegration degree of starch films and biocomposite films.

Sample	Disintegration Degree (%)
PS	91.2 ± 0.4
CS	92.3 ± 2.1
BM4	85.4 ± 2.3
BM8	80.8 ± 1.4
BM12	71.9 ± 1.3
CBM4	87.0 ± 1.3
CBM8	82.1 ± 3.3
CBM12	85.3 ± 1.2

## Data Availability

The data presented in this study are available upon request from the corresponding authors M.G.L.-R and R.M.-G.

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
