# Peer review of "Biodegradable Biocomposite of Starch Films Cross-Linked with Polyethylene Glycol Diglycidyl Ether and Reinforced by Microfibrillated Cellulose"

_polymers, 2024, doi:10.3390/polym16091290_

Round 1
Reviewer 1 Report
Comments and Suggestions for Authors
The experimental article “Biodegradable biocomposite of starch films cross-linked with PEGDE and reinforced by OCC microfibrillated cellulose” is devoted to the production of biocomposite films using: corn starch, a chemically safe glycerin plasticizer and a cross-linking agent, polyethylene glycol diglycidyl ether (PEGDE), and a reinforcing material - microfibrillated cellulose ( MFC) separated from old cardboard (OCC) sheets. The results of this study showed that PEGDE crosslinked plasticized starch films with 12 wt% OCC MFC exhibited superior tensile strength and tensile modulus results, as well as improved water absorption resistance of up to 19% compared to the reference PS material. To study the films obtained, the authors used modern instrumental methods: FTIR, NMR, XRD. It was also found that more than 80% degradation of the biocomposite was achieved in 15 days. The positive side of this article is the good quality of the illustrative material, as well as a comparison of our own results with the results of other researchers. Structurally, the article is presented logically, the course of research and discussion of the results are described in detail. The authors are recommended to increase relevance, since out of 65 literature sources used, only 8 of them are for 2021-2024), and also to adjust the title of the article. To publish a manuscript, it is recommended to eliminate the comments and follow the recommendations given in the list.
Notes and recommendations:
1. Correct the title of the article: remove or replace the abbreviations: the chemical compound “PEGDE” and the source of cellulose – “OCC”.
2. Abstract. Delete the term “nanofiber” due to the use of microfibers in this work. It is stated to be a solution casting method. The materials and methods show that, most likely, this is not a solution, but a suspension. It is recommended to harmonize the text in the annotation.
3. Introduction. Add or replace, if possible, links to the topic of the article with the release of 2023-2024.
4. Introduction. Lines 50-51. The authors' claim that microfibrillated cellulose (MFC) is identical to nanofibrillated cellulose (NFC) is controversial. It is recommended to rewrite this statement in a more reasonable manner.
5. Methods. Section 2.2.1. Line 90-92. The authors need to provide in more detail the conditions for alkaline processing of cellulose, modes of mechanical fibrillation of cellulose, and indicate what kind of product was obtained: MFC or NFC?
6. Methods. Section 2.2.2. Line 96. It is necessary to justify the choice of concentration of PEGDE and Glycerol used to obtain biocomposite films.
7. Methods. It is recommended to justify the choice of MFC additive range.
8. Results and their discussion. Lines 256-259. The text contains a description of nanofibers. It is difficult to estimate their mass content. Most likely, MFCs provide the positive properties of composite films. It is recommended that the authors do not insist that NFCs were obtained through the microfibrillation process.
9. Results and their discussion. Section 3.3.3. X-Ray Diffraction Analysis (XRD). It is recommended to provide the XRD for MFC and discuss crystallinity in the native form.
10. Results and their discussion. Lines 422-425. The authors make two contradictory statements in one sentence: “The results found in this study suggest that the cross-linking of starch and the incorporation of reinforcement do not interfere with the biodegradation of the materials; instead, they delay the time in which this effect takes place, thus reducing the rate of biodegradation, which depending on the applications could be considered an advantage.” It is necessary to rewrite so as not to mislead readers.
Author Response
Reviewer 1
The experimental article “Biodegradable biocomposite of starch films cross-linked with PEGDE and reinforced by OCC microfibrillated cellulose” is devoted to the production of biocomposite films using: corn starch, a chemically safe glycerin plasticizer and a cross-linking agent, polyethylene glycol diglycidyl ether (PEGDE), and a reinforcing material - microfibrillated cellulose ( MFC) separated from old cardboard (OCC) sheets. The results of this study showed that PEGDE crosslinked plasticized starch films with 12 wt% OCC MFC exhibited superior tensile strength and tensile modulus results, as well as improved water absorption resistance of up to 19% compared to the reference PS material. To study the films obtained, the authors used modern instrumental methods: FTIR, NMR, XRD. It was also found that more than 80% degradation of the biocomposite was achieved in 15 days. The positive side of this article is the good quality of the illustrative material, as well as a comparison of our own results with the results of other researchers. Structurally, the article is presented logically, the course of research and discussion of the results are described in detail. The authors are recommended to increase relevance, since out of 65 literature sources used, only 8 of them are for 2021-2024), and also to adjust the title of the article. To publish a manuscript, it is recommended to eliminate the comments and follow the recommendations given in the list.
Response: The authors acknowledge the reviewer's work conducted by comments and suggestions to improve the quality and clarity of our manuscript. Therefore, some references were replaced by updated versions and all recommendations were addressed (in the corrected version these changes are highlighted in green color).
Notes and recommendations:
1.Correct the title of the article: remove or replace the abbreviations: the chemical compound “PEGDE” and the source of cellulose – “OCC”.
Response: We agree with reviewer and title was modified to “Biodegradable biocomposite of starch films cross-linked with polyethylene glycol diglycidyl ether and reinforced by microfibrillated cellulose”
- Abstract. Delete the term “nanofiber” due to the use of microfibers in this work. It is stated to be a solution casting method. The materials and methods show that, most likely, this is not a solution, but a suspension. It is recommended to harmonize the text in the annotation.
Response: We agree. The term “nanofiber” was changed to “microfiber” and “solution” was changed to “suspension”.
- Introduction. Add or replace, if possible, links to the topic of the article with the release of 2023-2024.
Response: We agree and some references were replaced by:
- Schutz, G.F.; de Ávila, G.S.; Varcelino, A.R.M. Vieira, R.P. A review of starch-based biocomposites reinforced with plant fibers. Int. J. Biol. Macromol. 2024, 261, 129916.
- Diaz-Baca, J.A.; Fatehi, P. Production and characterization of starch-lignin based materials: A review. Biotechnol. Adv. 2024, 70, 108281.
- Fatima, S.; Khan, M.R.; Ahmad, I.; Sadiq, M.B. Recent advances in modified starch based biodegradable food packaging: A review. Heliyon. 2024, 10, e27453.
- Dutta, D.; Sit, N. Comprehensive review on developments in starch-based films along with active ingredients for sus-tainable food packaging. Sustain. Chem. Pharm. 2024, 39, 101534.
- Wu, H.; Lei, Y.; Lu, J.; Zhu, R.; Xiao, D.; Jiao, C.; Xig, R.; Zhang, Z.; Shen, G.; Liu, Y.; Li, S.; Li, M. Effect of citric acid induced crosslinking on the structure and properties of potato starch/chitosan composite films. Food Hydrocoll. 2019, 97, 105208.
Replaced references:
- Lomelí-Ramírez, M.G. Development of biocomposites of thermoplastic starch reinforced with green coconut fiber. Doctoral thesis in Forestry Engineering. Federal University of Paraná, Brazil, 2011.
- Ma, X.; Yu, J.; Kennedy, J.F. Studies on the properties of natural fibers-reinforced thermoplastic starch composites. Carbohydr. Polym. 2005, 62(1), 19-24.
- Shanks, R.; Kong, I. Thermoplastic Starch. In Thermoplastic Elastomers, 1rst ed.; El-Sonbati, A., Ed.; InTech; 2012, Volume 1, pp. 95-115.
- Peinado, M.D. Study of the biodegradability and disintegration of starch and PVA-based films incorporating different antimicrobial substances. Thesis for obtaining a degree in Food Science and Technology, Universitat Politècnica de València, Spain, 2015.
Although there are some other references under the year 2023 in this section, the authors consider them as relevant support for the document.
- Introduction. Lines 50-51. The authors' claim that microfibrillated cellulose (MFC) is identical to nanofibrillated cellulose (NFC) is controversial. It is recommended to rewrite this statement in a more reasonable manner.
Response: We agree with the reviewer's suggestion and MFC replaced the term NFC in the document to avoid reader confusion.
- Methods. Section 2.2.1. Line 90-92. The authors need to provide in more detail the conditions for alkaline processing of cellulose, modes of mechanical fibrillation of cellulose, and indicate what kind of product was obtained: MFC or NFC?
Response: In the corrected version, some more data were included in this part according to the main points of the methodology. Furthermore, for more details about this methodology, they can be consulted in our previous work (ref 25):
Line 92-97: The alkaline pulping process was carried out using a Jayme digester, where native OCC fibers were combined with the pulping liquor consisting of 10% NaOH and 0.1% anthra-quinone. The production of the MFCs was carried out through a mechanical fibrillation process using a Super Mascolloider Microprocessor colloidal mill (Masuko Sangyo MKCA6-2). Further information on this methodology, can be consulted in the work of González et al. [25].
- Methods. Section 2.2.2. Line 96. It is necessary to justify the choice of concentration of PEGDE and Glycerol used to obtain biocomposite films.
Response: We agree with the reviewer's suggestion and the next paragraph was included:
Line 99-103: The ratio of glycerol, cross-linker agent and reinforcing fibers in the starch was preliminary tested to determine the appropriate concentration of each component to obtain bio-composite films with optimal physical-mechanical properties. Thus, biocomposites were developed in film form from starch plasticized with 30% glycerol (PS), cross-linked using 3% PEGDE (CS),…
- Methods. It is recommended to justify the choice of MFC additive range.
Response: This point was also included in paragraph of comment 6 (Line 99-103).
- Results and their discussion. Lines 256-259. The text contains a description of nanofibers. It is difficult to estimate their mass content. Most likely, MFCs provide the positive properties of composite films. It is recommended that the authors do not insist that NFCs were obtained through the microfibrillation process.
Response: We agree with the suggestion and the term NFC was removed to avoid confusion for the reader.
- Results and their discussion. Section 3.3.3. X-Ray Diffraction Analysis (XRD). It is recommended to provide the XRD for MFC and discuss crystallinity in the native form.
Response: We agree and data included in the next paragraph:
Line 327-329: The crystallinity of native fibers and MFC was documented in a previous publication [25]. The calculated relative crystallinity index (Cr.I.) showed values ranging from 80 to 85, with the highest values observed in the microfibrillated cellulose.
- Results and their discussion. Lines 422-425. The authors make two contradictory statements in one sentence: “The results found in this study suggest that the cross-linking of starch and the incorporation of reinforcement do not interfere with the biodegradation of the materials; instead, they delay the time in which this effect takes place, thus reducing the rate of biodegradation, which depending on the applications could be considered an advantage.” It is necessary to rewrite so as not to mislead readers.
Response: We agree and the paragraph was modified to avoid confusion:
Line 429-431: The results found in this study suggest that the cross-linking of starch and the incorporation of reinforcement do not interfere with the biodegradation of the materials.

Reviewer 2 Report
Comments and Suggestions for Authors
a very interesting topic important from an ecological level. I would argue that it is too far-fetched to say that starch is a cheap raw material... maybe it would be better to stick to the available one. Because based on the research undertaken and, in fact, the desire to protect the environment, the technology for obtaining starch is not low-emission and waste-free.
chapter 2.2.3 and 2.2.5. - should be divided into subchapters
Moisture absorption. - there is no basic information about what mathematical model the data is calculated with, this information is only found in the presentation and discussion of the results. See why Langmuir's theories were chosen. On what basis do the authors assume that the adsorption phenomenon takes place as a monolayer?
the work lacked statistical analysis.
Table 2 - are the analyzed values statistically significant? If so, at what level?
Author Response
Reviewer 2
The authors acknowledge the reviewer's work conducted by comments and suggestions to improve the quality and clarity of our manuscript. Therefore, all suggestions were addressed (in the corrected version these changes are highlighted in yellow color).
a very interesting topic important from an ecological level. I would argue that it is too far-fetched to say that starch is a cheap raw material... maybe it would be better to stick to the available one. Because based on the research undertaken and, in fact, the desire to protect the environment, the technology for obtaining starch is not low-emission and waste-free.
Response: We agree with reviewer´s suggestion and “Low cost” was removed from the text..(high availability was already stated).
chapter 2.2.3 and 2.2.5. - should be divided into subchapters
Response: Thanks for the suggestions. However, we decided better modify the titles in 2.2.3 and 2.2.5 (to include all experiments) and avoid increasing more subsections.
2.2.3. Physical-Mechanical Performance and Water Adsorption Evaluation
2.2.5. Chemical and Structural Characterization
Moisture absorption. - there is no basic information about what mathematical model the data is calculated with, this information is only found in the presentation and discussion of the results. See why Langmuir's theories were chosen. On what basis do the authors assume that the adsorption phenomenon takes place as a monolayer?
Response: The diffusion coefficients (D) were determined from moisture uptake data using a hindered diffusion model proposed by Carter and Kibler [reference 38], which resembles a Langmuir-type model for anomalous moisture diffusion. This model, successfully applied to various composite materials [references 38,39], links absorption to free-volume availability and polymer–water affinity. Table 3 displays the parameters calculated via non-linear regression using Matlab® (this information was included in the corrected version: Line 214-225).
the work lacked statistical analysis.
Table 2 - are the analyzed values statistically significant? If so, at what level?
Response: We agree with the reviewer´s comment and data of the analysis of variance (ANOVA) and Tukey test (p < .05) were carried out to determine significant differences. Included in table 2.
